# Impact of Multidisciplinary Stroke Post-Acute Care on Cost and Functional Status: A Prospective Study Based on Propensity Score Matching

**DOI:** 10.3390/brainsci11020161

**Published:** 2021-01-26

**Authors:** Chong-Chi Chiu, Jhi-Joung Wang, Chao-Ming Hung, Hsiu-Fen Lin, Hong-Hsi Hsien, Kuo-Wei Hung, Herng-Chia Chiu, Shu-Chuan Jennifer Yeh, Hon-Yi Shi

**Affiliations:** 1Department of General Surgery, E-Da Cancer Hospital, Kaohsiung 82445, Taiwan; chiuchongchi@gmail.com (C.-C.C.); ed100647@edah.org.tw (C.-M.H.); 2School of Medicine, College of Medicine, I-Shou University, Kaohsiung 82445, Taiwan; 3Department of General Surgery, Chi Mei Medical Center, Liouying 73657, Taiwan; 4Department of Medical Research, Chi Mei Medical Center, Tainan 71004, Taiwan; 400002@mail.chimei.org.tw; 5Department of Neurology, Kaohsiung Medical University Hospital, Kaohsiung 80756, Taiwan; sflin@kmu.edu.tw; 6Department of Internal Medicine, St. Joseph Hospital, Kaohsiung 80288, Taiwan; gavinhsien@gmail.com; 7Division of Neurology, Department of Internal Medicine, Yuan’s General Hospital, Kaohsiung 80249, Taiwan; dimondwave@gmail.com; 8Department of Healthcare Administration and Medical Informatics, Kaohsiung Medical University, Kaohsiung 80708, Taiwan; chiu@kmu.edu.tw (H.-C.C.); syeh@faculty.nsysu.edu.tw (S.-C.J.Y.); 9Institute of Hospital Management, Tsinghua University, Beijing 100084, China; 10Department of Business Management, National Sun Yat-sen University, Kaohsiung 80424, Taiwan; 11Department of Medical Research, Kaohsiung Medical University Hospital, Kaohsiung 80756, Taiwan; 12Department of Medical Research, China Medical University Hospital, China Medical University, Taichung 40402, Taiwan

**Keywords:** stroke, post-acute care, cost, functional status

## Abstract

Few papers discuss how the economic burden of patients with stroke receiving rehabilitation courses is related to post-acute care (PAC) programs. This is the first study to explore the economic burden of stroke patients receiving PAC rehabilitation and to evaluate the impact of multidisciplinary PAC programs on cost and functional status simultaneously. A total of 910 patients with stroke between March 2014 and October 2018 were separated into a PAC group (at two medical centers) and a non-PAC group (at three regional hospitals and one district hospital) by using propensity score matching (1:1). A cost–illness approach was employed to identify the cost categories for analysis in this study according to various perspectives. Total direct medical cost in the per-diem-based PAC cohort was statistically lower than that in the fee-for-service-based non-PAC cohort (*p* < 0.001) and annual per-patient economic burden of stroke patients receiving PAC rehabilitation is approximately US $354.3 million (in 2019, NT $30.5 = US $1). Additionally, the PAC cohort had statistical improvement in functional status vis-à-vis the non-PAC cohort and total score of each functional status before rehabilitation and was also statistically significant with its total score after one-year rehabilitation training (*p* < 0.001). Early stroke rehabilitation is important for restoring health, confidence, and safe-care abilities in these patients. Compared to the current stroke rehabilitation system, PAC rehabilitation shortened the waiting time for transfer to the rehabilitation ward and it was indicated as an efficient policy for treatment of stroke in saving medical cost and improving functional status.

## 1. Introduction

According to the latest global data from the World Health Organization (WHO), stroke is now the second-leading cause of death [1]. In Taiwan, data collected by the National Health Interview Survey revealed that the age-standardized incidence of a first-ever stroke reached as high as 320 per 100,000 population in 2012 [2]. In most developed countries, post-acute care (PAC) and long-term care systems provide support or back-up support for patients with stroke [3]. In the United States of America, for example, up to 62.6–74.5% patients with stroke undergo PAC rehabilitation after hospital discharge [4,5]. In Taiwan, stroke is the disease most likely to require a prolonged hospital stay, and up to 10.4% of stroke patients require a prolonged hospital stay in an acute care ward [6]. Hospital stays for stroke are prolonged in Taiwan for two reasons: the low copayment for hospitalization and the lack of long-term care systems [6,7].

In Taiwan, a Post-Acute Care–Cerebrovascular Diseases (PAC-CVD) program was implemented in 2014 to improve resource allocation and patient outcomes by transferring patients in post-acute phase of stroke from medical centers to community hospitals, including regional and district hospitals [3]. However, few studies have discussed the roles of PAC programs in the economic burden of stroke rehabilitation and in functional recovery in patients with stroke [8,9,10]. This study is, to our knowledge, the first to apply propensity score matching (PSM) in a natural experimental design to examine the longitudinal impacts of stroke PAC programs on medical utilization and functional status. 

Therefore, this cohort study prospectively purposed to explore the economic burden of patients with stroke receiving post-acute care (PAC) rehabilitation and to explore the impact of multidisciplinary PAC programs on medical cost and functional status among patients with stroke.

## 2. Materials and Methods

### 2.1. The PAC Program

Multidisciplinary PAC stroke teams in Taiwan include neurologists, physiatrists, physiotherapists, occupational therapists, speech therapists, and nurses. The PAC rehabilitation program, which is prescribed by the physiatrist, comprises a complex program of universal activities performed at least three times per day. Depending on the condition of the patient, the components of the rehabilitation program may include facilitation, passive range of motion exercise, therapeutic exercise, bed mobility training, balance training, functional electric stimulation, training under suspension, ambulation training, transfer training, activities of daily living training, functional training, coordination training, posture training, speech training, and swallowing training. Hospitals receive a packaged and function-related reimbursement per day of rehabilitation. The maximal packaged reimbursement is NT $117.6 per day for high-intensity rehabilitation and NT $79.0 per day for general-intensity rehabilitation. The reimbursement covers all medical expenses for stroke care, including management of associated comorbidities/complications and stroke rehabilitation. 

The traditional rehabilitation program received by the non-PAC cohort was identical to that received by the PAC cohort; however, physical therapy, occupational therapy, and speech/swallowing therapy were limited to once per day in the non-PAC cohort. Notably, the fiscal incentive for a medical center to transfer a patient to a regional or a district hospital is mitigated by several factors, including the PAC-CVD transfer policy, the willingness of the stroke patient (or family) to accept responsibility for PAC care, and whether the physician agrees to the transfer. These mitigating factors should be considered by health providers when setting policies for hospital stays after stroke. Health providers should also be mindful that a short length of stay (LOS) after stroke does not necessarily indicate a good outcome [11]. Another difference between this PAC-CVD program and the traditional program is that the reimbursement for inpatient stroke rehabilitation in the PAC-CVD program was on a per-diem basis whereas reimbursement for the traditional stroke rehabilitation program was on a fee-for-service basis.

### 2.2. Study Design and Sample

The study population included all patients who had undergone stroke (defined as ICD-9-CM-codes: 433.x, 434.x, and 436.x for ischemic stroke; 430 and 431 for hemorrhagic stroke) admission to a PAC ward at two medical centers and a non-PAC ward at three regional hospitals and one district hospital in south Taiwan between March 2014 and October 2018. The patients should meet the criteria including acute stroke, stroke onset day within 30 days, and Modified Rankin Scale (MRS) scores of 2 to 4, where MRS scores of 0, 1, 2, 3, 4, and 5 are interpreted as no symptoms, no significant disability, slight disability, moderate disability, moderately severe disability, and severe disability, respectively [12]. 

Potential selection bias was further minimized by using a PSM approach to select stroke patients in the PAC cohort used for comparisons. The present study used the caliper matching method (also known as the greedy algorithm) with 1 to 1 match between the PAC cohort (at two medical centers) and the non-PAC cohort (at three regional hospitals and one district hospital) based on the propensity score [13]. Thus, the PAC cohort of all participants with 455 patients yielded 455 patients in the non-PAC cohort for statistical analysis (Figure 1). These stroke patients completed the pre-rehabilitation and the first year after rehabilitation assessments in this study after providing written informed consent. The study protocol was approved by the local ethic committee of the Kaohsiung Medical University Hospital (KMUH-IRB-20140308). Participants gave written informed consent before data collection began.

### 2.3. Estimation of Cost

The cost analysis was performed from the hospital perspective, implying that direct medical costs were included in the cost analysis. Direct medical costs were costs of medical treatments for stroke patients in Taiwan. Cost items of the direct medical cost included diagnosis fee, ward fee, examination fee, medicine and pharmacy service fee, rehabilitation therapy fee, and others. In order to facilitate comparisons with other economic evaluations, all cost inputs were inflated to 2020 US dollars. All costs were discounted annually by 3%.

### 2.4. Instruments of Functional Status and Chart Review

The functional status measures included the Barthal Index (BI), Functional Oral Intake Scale (FOIS), Mini-Mental State Examination (MMSE), Instrumental Activities of Daily Living Scale (IADL), EuroQoL Quality of Life Scale (EQ5D), and Berg Balance Scale (BBS) and the Chinese version of these measures has been validated and extensively used in both clinical practice and health research [3,14]. 

The following patient data obtained by records review and questionnaire interview were tested as independent variables in this study: gender, age, Charlson Comorbidity Index (CCI), type of stroke, comorbid diseases (e.g., diabetes, hypertension, hyperlipidemia, and coronary artery disease), stroke history, tissue plasminogen activator (tPA), thrombectomy, medications (e.g., statins, antiplatelets, anticoagulants, and antihypertensives), acute care LOS, LOS during rehabilitation, and total score for each functional status measure before rehabilitation.

### 2.5. Statistical Analysis

The unit of analysis in this study was the individual stroke patient. Descriptive statistics were tabulated to depict the stroke patient demographics. Regarding total direct medical cost at hospitalization, the standard administrative claims data required by the Taiwan Bureau of National Health Insurance (BNHI) include fees for the following: physician, radiology, physical therapy, hospital room, pharmacy, laboratory, special materials, and others. Incidence-based approach was employed in estimating the per-patient economic burden of total direct medical cost in the PAC cohort and non-PAC cohort during year one after rehabilitation. To reflect changes in real dollar value, all dollar values were converted to their equivalent 2020 values; New Taiwan Dollar (NTD) values were then converted to USD values at the average exchange rate over the four-year period of 2015–2018 (in 2020, NT $30.5 = US $1).

Effect size (ES) was calculated to directly compare the relative magnitude of change as measured by the two functional status measures. Thus, ES was calculated as the difference between the mean scores for two time intervals divided by the standard deviation of the score for the previous time interval [15]. Using this method of standardizing the extent of change measured by an instrument enabled comparisons between two instruments. An ES of 1.0 is equivalent to a change of one standard deviation in the sample. Effect sizes of 0.2, 0.5, and 0.8 are typically considered small, medium, and large changes, respectively. Differences in ES and associated 95% confidence intervals are also employed using bias-corrected and accelerated bootstrapping with 1000 replications [16]. Additionally, multiple linear regression analysis was carried out to determine the significant predictors of total direct medical costs and total score of each functional status measure of stroke patients after one-year rehabilitation. Log transformation was undertaken for total direct medical costs to reduce the skewness and number of outliers, and to improve normality, linearity, and homoscedasticity of residuals.

Standard regression diagnostics were used to check the assumptions for the regression analyses [17]. Statistical analyses were performed using Stata Statistical Package, version 13.0 (StataCorp LP, College Station, TX, USA). All tests were two-sided, and *p* values less than 0.05 were considered statistically significant.

## 3. Results

Table 1 compares the PAC cohort and the non-PAC cohort in terms of patient characteristics, and there were no statistically significant differences between these two cohorts after PSM. Most of the patients suffered from ischemic stroke, and among the common risk factors of stroke, over 50% of patients had a history of hypertension, hyperlipidemia, and diabetes mellitus. Furthermore, the annual total direct medical cost in the PAC cohort (per-diem-based LOS, mean US $5326.7, standard deviation (SD) US $1933.5) was significantly lower than that in the non-PAC cohort (fee-for-service-based LOS, mean US $10175.8, SD US $2377.9) (*p* < 0.001) (Table 2). The per-patient annual economic burden of total direct medical cost of stroke patients receiving PAC programs after rehabilitation is approximately to US $−354.6 million.

The effect size (ES) for total score of each functional status measure is shown as between 0.85 and −0.43 in PAC cohort and 0.04 to 0.71 in non-PAC cohort (Table 3). During the study period, both in the PAC cohort and non-PAC cohort, the IADL showed the largest improvement, but the EQ5D had least improvement. Furthermore, in comparison with the non-PAC cohort, the PAC cohort showed significant improvement among all functional status measures (*p* < 0.05). 

In Table 4, after adjusting for effective predictors, total direct medical cost in the PAC cohort was significantly lower than that in the non-PAC cohort (*p* < 0.001) and it is also shown that the PAC cohort had statistical improvement in functional status vis-à-vis the non-PAC cohort, and total score of each functional status before rehabilitation was also statistically significant with its total score after rehabilitation training (*p* < 0.001).

## 4. Discussion

Costs for stroke are related to the risk factors (such as hypertension, hyperlipidemia, diabetes mellitus, atrial fibrillation, and previous stroke history) and the social-economic status [9,10]. Without insurance coverage, managing stroke constitutes a huge direct-cost burden generally unaffordable by a stroke sufferer. Fattore et al. classified the cost into three categories as direct healthcare, direct non-healthcare (informal care costs and paid care costs), and productivity losses [18]. The major three cost components were informal care cost, initial hospitalization cost, and rehabilitation cost ranked in order. Moreover, to investigate the economic burden of stroke patients in the PAC rehabilitation program, this study utilized the 2018 data from BNHI to calculate the incidence of stroke. Additionally, health insurance claims data were used to estimate total direct medical cost. Taiwan has a single, national health insurance system; therefore, the national incidence of stroke could be estimated. This method has the added advantage of maintaining internal consistency. Although we used national health insurance claims data, the incidence rates reported here were similar to data from stroke registries. 

A more intensive PAC program was incorporated into the Japanese medical insurance system after 2000. In Miyai’s study [19], the dose-dependent effect of hours of therapy on rehabilitation outcome after stroke was obvious. These patients received higher discharge motor and function levels with a shorter LOS. Under the current National Health Insurance system in Taiwan, all patients are limited to receive physical therapy, occupational therapy, and speech therapy once per day. In our study, patients received more rehabilitation programs to their level of tolerance. Patients could regain various functions earlier and complications were prevented or minimized. The per-diem reimbursement avoided the disadvantage of financial burden in fee-for-service reimbursement. Thus, the training program was efficient and economical. 

An organized, multidisciplinary team approach should be initiated early after the onset of acute stroke to minimize functional disability, prevent complications, and hence decrease prolonged hospital stays. According to the Get With The Guidelines–Stroke (GWTG-Stroke) program, among 616,982 adults with stroke diagnosis, almost 90% of them had documentation of an acute assessment for rehabilitation [20]. However, the inpatient stroke rehabilitation utilization in Taiwan was only 34.0% (33.0% for physical therapy, 19.6% for occupational therapy, and 5.3% for speech therapy), much lower than those observed in the United States, Canada, the United Kingdom, and Austria (59% to 75% for physical therapy, 16% to 39% for occupational therapy, and 10% to 23% for speech therapy) [7,21]. Furthermore, in some local hospitals, inpatient rehabilitation was carried out in the form of bedside programs, without rehabilitation facilities. It was noted that some stroke rehabilitation therapists in some local hospitals lacked experience; thus, it was necessary to qualify and classify the stroke rehabilitation provider by adjusting the payment system. Our study results revealed that early intensive stroke rehabilitation led to cost-effective and efficient outcomes. The multidisciplinary team approach was also important in achieving the outcomes.

A recently published study addressed the impact of female gender on stroke subtype, risk factors, severity, and outcome [22]. The authors concluded that Fragile X syndrome is a genetic condition known to increase the risk of cognitive impairment and socio-emotional challenges in affected males and females [23]. Another study by Arboix et al. reported that, compared to men with acute stroke, women with acute stroke have a significantly higher risk of death in the immediate post-stroke phase (13.5% vs. 10.8% in males; *p* = 0.006) and a significantly lower probability of early full neurological recovery (11.8% vs. 13.9% in males; *p* = 0.029) [22]. Theoretically, stroke should tend to occur at an older age in women compared to men. Since women are more likely to have cardioembolism-related risk factors, they are also more likely to have the cardioembolic stroke subtype, which may explain the poorer stroke outcomes reported in women compared to men [22,23]. Over the 24-year period of the current study, trends observed in women relative to men included older age, lower mortality, shorter hospitalization, and higher incidences of hypertension, atrial fibrillation, and cardioembolic infarction. In Kuptniratsaikul’s article review, the average LOS for inpatient rehabilitation after stroke was about a month, except for the USA (29.4 days in Thailand, 31.3 days in Ireland, 31.2 days in Switzerland, 37.1 days in Singapore, and 21.9 days in Texas) [24]. High economic burden from stroke was the most important reason for the shorter average LOS in the US. Few studies have discussed the inpatient rehabilitation after acute stroke [3,25]. The PAC programs shortened the LOS, especially shortening the days in waiting to transfer to the rehabilitation ward and reducing the readmission rates. Additionally, patients were transferred to PAC rehabilitation units as early as possible to maximize rehabilitation efficacy, to maximize the potential for functional restoration, and to minimize costs [3,25]. Early and intensive physical therapy, occupational therapy, and speech therapy are also important for successful rehabilitation of stroke patients. A collaborative effort by a multidisciplinary team was another important contributor to good outcomes. The Taiwan NHI system ensures that all hospitals that provide PAC receive the same reimbursement, regardless of care quality or hospital accreditation level.

In the study, this inpatient stroke rehabilitation program had four major differences from the current implementation policy. Firstly, the reimbursement was per diem but not fee-for-service. Secondly, patients could receive more intensive and more frequent rehabilitation programs. Thirdly, every patient should have his or her functional status re-evaluated every three weeks and these medical records should be sent to the NHI system. Fourthly, no matter the hospital accreditation level (medical center, regional hospital, or district hospital), the payment was all the same. 

In Taiwan, stroke patients with prolonged hospital stay constitute only 10.4% of the total stroke patients but 38.9% of the total person–hospital bed days and 47.8% of the total in-hospital medical expenses [6]. Besides surgical intervention and mechanical ventilation use, rehabilitation need for physical/ADL dependency and speech/swallowing problems is a major cause to delay discharge from the hospital [26,27,28]. If stroke patients were referred successfully, the problems concerning prolonged hospital stay in medical centers would be resolved; however, a few stroke patients are transferred from medical centers to PAC program hospitals in the whole country. The major reasons why transfers from medical centers are limited include the following: stroke patients and their family have low confidence in local hospitals; some neurologists and neurosurgeons have limited knowledge of PAC programs; some local hospitals are not well prepared; and people treated for stroke in medical centers often insist on waiting for transfer to rehabilitation wards.

Although all research questions were adequately and satisfactorily addressed, three limitations are noted. Firstly, this study collected data from acute stroke patients from the stroke onset day within 30 days in two district hospitals, having one of the highest number of PAC-program stroke patients in Taiwan. Such a sample selection ensures that the limited experience of physicians and medical professionals does not significantly influence patient outcomes. Secondly, the etiology of ischemic stroke affects prognosis, outcome, and management. Trials of acute stroke therapies should investigate how responses are influenced by ischemic stroke subtype. The current study did not consider differences in ischemic stroke subtype, stroke risk, bleeding risk (i.e., CHA2DS2-VASc score and HAS-BLED score), or medications (i.e., statins, antiplatelets, anticoagulants, and antihypertensives). Additionally, we acknowledge that more data and longer survey time periods are needed to understand the complex patterns that emerged in the current analysis.

In conclusion, improvements in mobility function were larger than improvements in emotional and mental function. Early PAC rehabilitation is important for restoring health, confidence, and safe-care abilities in these patients. Compared to the current stroke rehabilitation system, PAC rehabilitation achieved larger improvements in functional status while simultaneously reducing waiting time for transfer to rehabilitation ward, total LOS, and total direct medical costs. The PAC rehabilitation was effective training for rehabilitation of stroke patients. This study is the first to discuss the cost-effectiveness of intensive post-stroke rehabilitation under a per-diem payment system.

## Figures and Tables

**Figure 1 brainsci-11-00161-f001:**
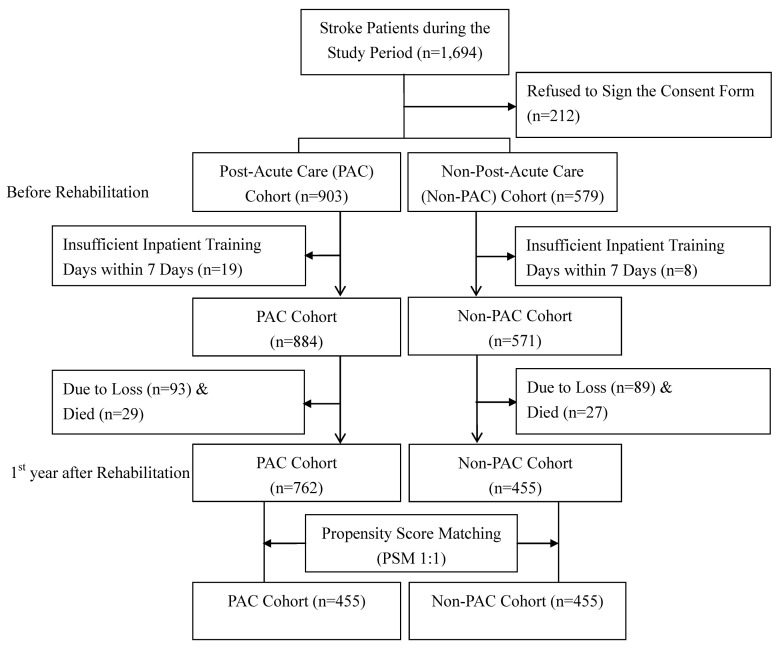
Flow chart of recruitment and study procedure.

**Table 1 brainsci-11-00161-t001:** Stroke patient characteristics.

		Before Propensity Score Matching			After Propensity Score Matching		
Variables		PAC Cohort (*n* = 903)	Non-PAC Cohort (*n* = 579)	*p* Value	PAC Cohort (*n* = 455)	Non-PAC Cohort (*n* = 455)	*p* Value
Age, years		61.9 ± 10.6	65.2 ± 11.9	0.152	65.0 ± 10.3	65.2 ± 10.7	0.934
Gender	Female	269 (29.8%)	261 (45.1%)	0.146	192 (42.3%)	190 (41.8%)	0.684
	Male	634 (70.2%)	318 (54.9%)		263 (57.7%)	265 (58.2%)	
Education, years		8.7 ± 4.2	9.0 ± 4.8	<0.001	8.8 ± 4.1	9.2 ± 4.3	0.888
BMI, kg/m^2^		24.1 ± 3.0	24.5 ± 3.8	0.968	24.2 ± 3.1	24.0 ± 3.4	0.949
Stroke type	Ischemic	769 (85.2%)	505 (87.2%)	0.674	387 (85.1%)	393 (86.3%)	0.910
	Hemorrhagic	134 (14.8%)	74 (12.8%)		68 (14.9%)	62 (13.7%)	
Hypertension	Yes	634 (70.2%)	420 (72.5%)	0.487	328 (72.1%)	329 (72.3%)	0.991
Hyperlipidemia	Yes	480 (53.2%)	340 (58.7%)	0.311	236 (51.8%)	240 (52.8%)	0.807
Diabetes mellitus	Yes	461 (51.0%)	306 (52.8%)	0.570	236 (51.8%)	239 (52.4%)	0.990
Atrial fibrillation	Yes	75 (8.3%)	48 (8.3%)	0.989	38 (8.3%)	37 (8.1%)	0.934
Previous stroke	Yes	117 (13.0%)	144 (24.9%)	<0.001	82 (18.0%)	84 (18.4%)	0.819
Acute care LOS, days		13.1 ± 27.3	24.5 ± 34.1	<0.001	23.5 ± 11.4	24.0 ± 11.6	0.369
LOS during rehabilitation, days		31.2 ± 17.5	37.0 ± 12.9	<0.001	35.2 ± 12.1	36.3 ± 11.4	0.816
BI score before rehabilitation		44.7 ± 23.5	46.0 ± 26.4	0.403	44.7 ± 20.2	46.0 ± 27.0	0.279
FOIS score before rehabilitation		5.9 ± 1.8	6.1 ± 2.0	0.310	5.9 ± 2.0	6.1 ± 2.1	0.974
EQ5D score before rehabilitation		9.6 ± 1.9	8.9 ±1.9	0.593	9.6 ± 1.9	8.8 ± 2.0	0.712
IADL score before rehabilitation		2.0 ± 1.3	2.3 ± 1.4	0.328	2.0 ± 1.1	3.1 ± 1.3	0.147
BBS score before rehabilitation		20.6 ± 16.3	18.4 ± 15.2	0.486	20.6 ± 14.8	18.4 ± 14.0	0.392
MMSE score before rehabilitation		20.8 ± 7.5	22.3 ± 6.9	0.213	21.8 ± 11.0	23.3 ± 9.2	0.259

BMI, body mass index; LOS, length of stay; PAC, post-acute care; BI, Barthel Index; FOIS, Functional Oral Intake Scale; EQ5D, EuroQoL Quality of Life Scale; IADL, Instrumental Activities of Daily Living Scale; BBS, Berg Balance Scale; MMSE, Mini-Mental State Examination. Values are expressed as mean ± standard deviation or *n* (%).

**Table 2 brainsci-11-00161-t002:** Annual economic burdens of total direct medical cost per patient in PAC and non-PAC cohorts before and after one-year rehabilitation.

Cost Components	PAC Cohort (*n* = 455) ^#^	Non-PAC Cohort (*n* = 455)	Differences (PAC–non-PAC)	Economic Burden *
	Mean ± SD	Mean ± SD		
Diagnosis fee	4139.5 ± 1798.1	1089.0 ± 157.3		
Ward fee		2882.2 ± 560.9		
Examination fee		1619.0 ± 373.9		
Medicine and pharmacy service fee		450.1 ± 25.1		
Rehabilitation therapy fee		1103.5 ± 273.6		
Other fees		1785.9 ± 430.9		
Total direct medical cost during rehabilitation	4139.5 ± 1798.1	8929.8 ± 1827.1	−4790.3 ± 1805.7	
Total direct medical cost after discharge	1187.2 ± 1148.6	1246.0 ± 1203.6	−58.8 ± 35.6	
Total direct medical cost	5326.7 ± 1933.5	10,175.8 ± 2377.9	−4849.1 ± 2685.7	−354,886,232.6

PAC, post-acute care; SD, standard deviation; ICER, incremental cost-effectiveness ratio. ^#^ Mean direct medical cost for the PAC cohort, hospitals will receive a packaged and function-related reimbursement by day, that is, a maximal packaged imbursement of US $117.6 per day for high-intensity rehabilitation or US $79.0 per day for general-intensity rehabilitation covering whole medical expenses for stroke care, managing associated comorbidities and complications, and rehabilitation. * Economic burden during 1 year after rehabilitation is US $4849.1 per patient × 318.2 patients per 100,000 person-year (age-standardized incidence of first-ever stroke) × 23,000,000 persons (Taiwan nationwide population). Therefore, annual per-patient economic burden of total direct medical cost approximately equals to US $354.6 million.

**Table 3 brainsci-11-00161-t003:** Effect size (ES) and mean difference for total score of each functional status measure: comparison of post-acute care (PAC) and non-PAC cohorts.

	PAC Cohort (*n* = 455)			Non-PAC Cohort (*n* = 455)			PAC–non-PAC
Measures	Before Rehabilitation	After 1-year Rehabilitation	ES of after Rehabilitation vs. before Rehabilitation	Before Rehabilitation	After 1-year Rehabilitation	ES of after Rehabilitation vs. before Rehabilitation	Mean Difference (Estimate [95% C.I.]) *
BI	44.7	64.0	0.82	46.0	63.3	0.65	0.17 (0.07, 0.27)
FOIS	5.9	6.2	0.17	6.1	6.1	0.05	0.12 (0.09, 0.15)
EQ5D	9.6	8.8	−0.43	8.8	8.8	0.04	−0.39 (−0.43, −0.34)
IADL	2.0	3.1	0.85	3.1	3.3	0.71	0.14 (0.11, 0.16)
BBS	20.6	29.9	0.57	18.4	24.9	0.42	0.13 (0.07, 0.19)
MMSE	21.8	24.0	0.29	23.3	24.3	0.15	0.14 (0.11, 0.17)

BI, Barthel Index; FOIS, Functional Oral Intake Scale; EQ5D, EuroQoL Quality of Life Scale; IADL, Instrumental Activities of Daily Living; BBS, Berg Balance Scale; MMSE, Mini-Mental State Examination. * Mean difference is presented as effect size (95% confidence intervals obtained by bootstrapping).

**Table 4 brainsci-11-00161-t004:** The coefficients of significant variables of multiple linear regression model of log of total direct medical cost and total score of each functional status measure after adjusting all effective predictors (*n* = 910).

Variables	Cost	BI	FOIS	EQ5D	IADL	BBS	MMSE
	Coefficient	Coefficient	Coefficient	Coefficient	Coefficient	Coefficient	Coefficient
Study cohort (PAC vs. non-PAC)	−0.41 ***	10.34 ***	1.24 ***	−1.44 ***	1.47 **	12.89 ***	7.17 **
BI score before rehabilitation	-	0.74 ***	-	-	-	-	-
FOIS score before rehabilitation	-	-	0.57 ***	-	-	-	-
EQ5D score before rehabilitation	-	-	-	0.40 ***	-	-	-
IADL score before rehabilitation	-	-	-	-	0.79 ***	-	-
BBS score before rehabilitation	-	-	-	-	-	0.71 ***	-
MMSE score before rehabilitation	-	-	-	-	-	-	0.74 ***

^#^ Adjusted factors included gender, age, Charlson Comorbidity Index (CCI), type of stroke, comorbid diseases (e.g., diabetes, hypertension, hyperlipidemia, and coronary artery disease), stroke history, tissue plasminogen activator (tPA), thrombectomy, medications (e.g., statins, antiplatelets, anticoagulants, and antihypertensives), acute care lengths of stay (LOS), and LOS during rehabilitation. * *p* < 0.05, ** *p* < 0.01, *** *p* < 0.001. PAC, post-acute care; BI, Barthel Index; FOIS, Functional Oral Intake Scale; EQ5D, EuroQoL Quality of Life Scale; IADL, Instrumental Activities of Daily Living; BBS, Berg Balance Scale; MMSE, Mini-Mental State Examination.

## Data Availability

The data presented in this study are available on request from the corresponding author.

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
