# Peer review of "Impact of Multidisciplinary Stroke Post-Acute Care on Cost and Functional Status: A Prospective Study Based on Propensity Score Matching"

_brainsci, 2021, doi:10.3390/brainsci11020161_

Round 1

Reviewer 1 Report

The rehabilitation in the acute phase of stroke is very important. In this study the authors explored the economic burden of stroke patients receiving post-acute care rehabilitation and evaluated the impact of multidisciplinary program on cost and functional status of them.

The study is constructed properly, subject is interesting, English in my opinion is good. 

In my opinion the paper can be published in Brain Sciences.

Author Response

The rehabilitation in the acute phase of stroke is very important. In this study the authors explored the economic burden of stroke patients receiving post-acute care rehabilitation and evaluated the impact of multidisciplinary program on cost and functional status of them.

The study is constructed properly, subject is interesting, English in my opinion is good.

Ans:

Thank you for your encouraging comments.

Reviewer 2 Report

In most of the developed countries, post-acute care (PAC) and long-term care systems contribute to the support back-up [1-3]. The proportion of the total expenditure on (53) health to gross domestic product is much lower in Taiwan than in the other developed (54) countries [4,5]. However, no paper was published and further rehabilitation for stroke is (55) limited.

I am not sure what line 55 means, please elaborate on what was not published or if some info was obtained from other communication

On lines 278-282 you discuss some of the idiosyncrasies in Taiwan regarding stroke care.  It would be helpful to include some information in the introduction regarding the health care system, stroke designation, triage to stroke centers, etc. and/or lack there of.

were referred successfully, the problems concerning prolonged hospital stay in medical (278) centers would be resolved; however, a few stroke patients are transferred from medical (279) centers to PAC program hospitals in the whole country. The major reasons why transfers (280) from medical centers are limited include the following: stroke patients and their family (281) have low confidence in local hospitals; some neurologists and neurosurgeons have lim- (282) ited knowledge of PAC programs; some local hospitals are not well prepared;

was there any difference from one PAC to the other or the non PAC hospitals.

Author Response

In most of the developed countries, post-acute care (PAC) and long-term care systems contribute to the support back-up [1-3]. The proportion of the total expenditure on (53) health to gross domestic product is much lower in Taiwan than in the other developed (54) countries [4,5]. However, no paper was published and further rehabilitation for stroke is (55) limited.

I am not sure what line 55 means, please elaborate on what was not published or if some info was obtained from other communication

Ans:

Thank you for your suggestion. As advised by the reviewer, the Introduction section of the revised manuscript discusses related works and clarifies the gap in the literature that was addressed in this study (lines 15-19, page 3). Thank you for your suggestion.

On lines 278-282 you discuss some of the idiosyncrasies in Taiwan regarding stroke care.  It would be helpful to include some information in the introduction regarding the health care system, stroke designation, triage to stroke centers, etc. and/or lack there of.

were referred successfully, the problems concerning prolonged hospital stay in medical (278) centers would be resolved; however, a few stroke patients are transferred from medical (279) centers to PAC program hospitals in the whole country. The major reasons why transfers (280) from medical centers are limited include the following: stroke patients and their family (281) have low confidence in local hospitals; some neurologists and neurosurgeons have lim- (282) ited knowledge of PAC programs; some local hospitals are not well prepared;

was there any difference from one PAC to the other or the non PAC hospitals.

Ans:

As advised, the Introduction section of the revised manuscript provides additional background information regarding the Taiwan healthcare system and protocols for designation of stroke type and treatment (lines 2-13, page 3). Specifically, differences between PAC and non-PAC for patients with stroke also are discussed in the PAC Program subsection in the revised manuscript (lines 12-19, page 4 & lines 1-19, page 5). Thank you. 

Reviewer 3 Report

The authors present a prospective, cohort study  with the purpose of exploring the economic burden of stroke patients receiving post-acute care (PAC) rehabilitation and exploring the impact of multidisciplinary PAC program on medical cost and functional status among patients with stroke. The results of their study revealed  that early intensive stroke rehabilitation  led to cost-effective and efficient outcomes. The multidisciplinary team approach was also important in achieving the results. The study is interesting, but some aspects of the manuscript may be improved taking into account the following points:

1.It would be interesting to know the different ischemic stroke subtypes in the study population.

2 One of the future lines of research in the study of early PAC rehabilitation would be the relationship with gender. A recently published study on the impact of female gender on the distribution of risk factors, stroke subtype, stroke severity, and outcome (Clin Neurol Neurosurg 2014 Dec;127:19-24). A comment of this important aspect should be included in the Discussion.

Author Response

The authors present a prospective, cohort study  with the purpose of exploring the economic burden of stroke patients receiving post-acute care (PAC) rehabilitation and exploring the impact of multidisciplinary PAC program on medical cost and functional status among patients with stroke. The results of their study revealed  that early intensive stroke rehabilitation  led to cost-effective and efficient outcomes. The multidisciplinary team approach was also important in achieving the results. The study is interesting, but some aspects of the manuscript may be improved taking into account the following points:

1.It would be interesting to know the different ischemic stroke subtypes in the study population.

Ans:

We agree with the reviewer that stroke subtype would be of interest to readers. Lack of differentiation by subtype is noted in the Limitations sections of the revised manuscript (lines 6-11, page 17). Thank you very much for your observations.

2 One of the future lines of research in the study of early PAC rehabilitation would be the relationship with gender. A recently published study on the impact of female gender on the distribution of risk factors, stroke subtype, stroke severity, and outcome (Clin Neurol Neurosurg 2014 Dec;127:19-24). A comment of this important aspect should be included in the Discussion.

Ans:

As advised by the reviewer, the Discussion section in the revised manuscript mentions reported gender differences in outcomes of early PAC rehabilitation for patients with stroke (lines 7-19, page 14). Thank you.